# FLR: Label-Mixture Regularization for Federated Learning with Noisy Labels

**Taehyeon Kim**                                                              *kimtaehyeon610@gmail.com*
*KAIST AI*
**Donggyu Kim**[*]                                                            *eaststar9979@gmail.com*
*MediPixel*
**Se-Young Yun**                                                             *yunseyoung@kaist.ac.kr*
*KAIST AI*

**Reviewed on OpenReview:** *https://openreview.net/forum?id=Z8A3HDgSOE*

## Abstract

Label noise in federated learning (FL) has garnered increasing attention due to the decentralized nature of FL, where data is collected from multiple clients with potentially different levels of label noise. This study introduces two pivotal contributions to this domain. First, we anatomize the memorization phenomenon in FL into server-side and client-side components, marking the first investigation into how these distinct forms of memorization impact learning. Second, to mitigate the memorization in FL, we present the Federated Label-mixture Regularization (FLR) strategy, a straightforward yet effective approach that employs regularization through pseudo labels generated by merging local and global model predictions. This method not only improves the accuracy of the global model in both i.i.d. and non-i.i.d. settings but also effectively counters the memorization of noisy labels. We empirically find that FLR aligns with and advances existing FL and noisy label mitigation methods over multiple datasets under various levels of data heterogeneity and label noise.

## 1 Introduction

The advent of large-scale datasets has propelled deep neural networks to remarkable achievements in fields such as computer vision, information retrieval, and language processing (Yao et al., 2021). However, these datasets often contain sensitive personal information, making conventional centralized learning impractical for applications such as person identification, financial services, and healthcare systems. Federated learning (FL) addresses privacy issues by allowing clients (e.g., edge devices) to collaborate with a central server (e.g., a service manager) without sharing their local data. Instead, clients update their local models using their private data, and the central server aggregates these updates to improve the global model. This procedure is repeated until convergence (FedAvg) (McMahan et al., 2017). FL enables on-device learning and avoids systematic privacy risks at the data level, making it particularly suitable for the computing devices such as phones, tablets, and autonomous driving systems (Kim & Yun, 2022; Kim et al., 2023).

In FL, multiple clients may have different levels of label noise (Xu et al., 2022) due to various factors, such as annotator skill, client bias, malfunctioning data collectors (Kim et al., 2018), or even malicious tampering with labels (Chen et al., 2020). For instance, in the healthcare industry, manually labeling patient records at each hospital is susceptible to corruption due to the complexity of medical terminology and the potential for annotator bias. Additionally, data entry mistakes or misunderstandings by medical professionals may introduce inconsistencies or errors into the original data (Xu et al., 2019; Karimi et al., 2020). Training with noisy labels can negatively impact over-parameterized neural networks, as they are prone to memorizing

---

[*] Work done while at KAIST AI

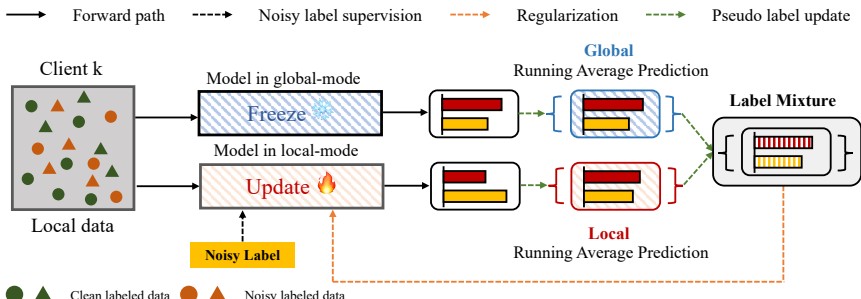

Figure 1: Overview of our proposed regularization, Federated Label-mixture Regularization (FLR).

mislabeled instances (a.k.a., memorization). Label noise, combined with FL's heterogeneous nature, can exacerbate this phenomenon. Traditional methods for addressing label noise in centralized learning (CL) are not directly applicable in the FL setting due to privacy constraints, small client datasets, and difficulties in handling data heterogeneity (Chen et al., 2020; Xu et al., 2022; Ji et al., 2024; Tsouvalas et al., 2024).

In this work, we first identifies and addresses the dual nature of memorization in FL – local memorization at the client level and global memorization at the server level (**Section 4**). We explore how local memorization, driven by client-specific data and biases, poses a significant risk of overfitting to noisy labels. Conversely, global memorization highlights the challenges faced by the central model in distilling accurate information from diverse, and potentially noisy, client updates. Our analysis reveals the intricate interplay between these two forms of memorization and their cumulative impact on the learning process in FL.

To combat these challenges, we introduce a simple yet efficient novel regularization for dealing with noisy labels in FL, termed as Federated Label-mixture Regularization (**FLR**; Figure 1), which supervises the noisy instances with pseudo labels generated through a combination of global server's running average predictions and local running average predictions. Our method is grounded in the recognition that memorization in FL manifests distinctively at server and client levels. In both i.i.d. and non-i.i.d. scenarios, the server's and client's temporal ensembling vectors each mitigate memorization issues. However, neither alone fully addresses both local and global memorization. This highlights the strength of our FLR method, which combines these vectors to effectively handle various noise conditions and data heterogeneity. Through experimental validation, we ascertain that leveraging a mixture of both vectors culminates in a robust label mixture adept at handling various noise conditions and data heterogeneity. Building upon this, we further develop several techniques by balancing the scale between global running average predictions and local running average predictions. In light of the above, our key contributions are as follows:

- We conduct exploratory experiments to analyze memorization patterns in FL, focusing on both client and server sides, to understand how label noise affects learning (**Section 4**).

- We demonstrate that FLR effectively prevents local and global memorization and outperforms other methods on benchmark datasets with various levels of data heterogeneity and label noise (**Section 4**, **Section 5**).

## 2 Related work

### 2.1 Learning with noisy label

Label noise in supervised learning can significantly reduce the generalization capability of deep neural networks (DNNs), leading to the development of various methods for robust training (Song et al., 2022). Recent progress in this field has included designing robust architectures (Yao et al., 2018; Momeny et al., 2021; Li et al., 2021), loss functions, and regularization techniques (Pereyra et al., 2017; Zhang et al., 2018; Liu et al., 2020; Wei et al., 2022; Kim et al., 2021b), as well as methods for loss adjustment (Reed et al., 2015; Ma et al., 2018;

Zheng et al., 2020) and sample selection (Han et al., 2018; Li et al., 2020a; Kim et al., 2021a). However, most of them have been proven to work only under CL settings.

One of the major challenges in dealing with label noise is the DNN's ability to memorize unreliable data, including random data or noisy labels, a.k.a., memorization phenomenon. This memorization phenomenon is observed to occur during the training phase, where DNNs first learn correctly labeled data and then gradually memorize the wrong labels (Zhang et al., 2017; 2021; Arpit et al., 2017). Bai et al. (2024) demonstrate that such mislabeled examples even can be memorized in the early stage. To prevent the memorization phenomenon, some approaches have focused on stopping training before memorization becomes severe (Song et al., 2019; Li et al., 2020b; Bai et al., 2021), and others have utilized regularization in the early-learning phase (Liu et al., 2020; Xia et al., 2020). However, their effectiveness in the FL setting has yet to be fully explored.

## 2.2 Federated learning with noisy label

Federated learning with noisy labels (FNL) is a scenario where the clients have label noise in their local datasets. This problem is particularly prevalent in FL, where data is decentralized and collected from multiple clients with varying levels of label noise. Training a model with noisy labels can be detrimental as over-parameterized neural networks are prone to fitting the training dataset, leading to a memorization problem. Existing FNL methods address this issue by using techniques such as sample selection, robust aggregation, label correction, and robust loss functions. For instance, FedCorr (Xu et al., 2022) corrects label noise in multiple stages, such as preprocessing, fine-tuning, and regular training, and FedRN (Kim et al., 2022) robustly aggregates local models by weighting them based on the estimated label noise level. FedNoRo (Wu et al., 2023) integrates noise detection via Gaussian models and robust updates through knowledge distillation and distance-aware aggregation. However, some studies have limitations, such as relying on supervision (Chen et al., 2020; Tuor et al., 2021) or not analyzing results in non-i.i.d. settings and situations with heterogeneous noise levels (Jiang et al., 2022; Kim et al., 2022; Ji et al., 2024; Zhou & Wang, 2024).

## 3 Preliminaries

The objective of FL is to solve the optimization problem for a distributed collection of heterogeneous data: $\min_{\boldsymbol{w}} f(\boldsymbol{w}) := \sum_{k \in S} \frac{n_k}{n} F_k(\boldsymbol{w})$ where $S$ is the set of total clients, with each client $k$ has $n_k$ local training data samples, and $n = \sum_{k \in S} n_k$. The local objective of client $k$ is to minimize $F_k(\boldsymbol{w}) = \mathbb{E}_{x_k \sim \mathcal{D}_k}[\ell_k(\boldsymbol{x}_k, \boldsymbol{y}_k; \boldsymbol{w})]$, where $\ell_k$ is the loss function parameterized by $\boldsymbol{w}$ on the local data $(\boldsymbol{x}_k, \boldsymbol{y}_k)$ from local data distribution $\mathcal{D}_k$. Here, we consider a setting having $N$ clients and dataset with $C$ classes. Each client $k$ has the training set $\{(\boldsymbol{x}_{k_i}, \boldsymbol{y}_{k_i})\}_{i=1}^{n_k}$, where $\boldsymbol{x}_{k_i}$ is the $i$th input and $\boldsymbol{y}_{k_i} \in \{0,1\}^C$ is the corresponding one-hot label vector; $\boldsymbol{y}_{k_i}^{(c)} = 1$ iff $\boldsymbol{x}_{k_i}$ belongs to class $c$. The main issue is that it is unknown whether such labels are correctly annotated (i.e., clean) or not (i.e., noisy).

**Data heterogeneity**  In order to consider the data heterogeneity in FL, we assume the true data distribution $\mathcal{D}_k$ prior to introducing label noise. In i.i.d. setting, all clients have the same size local training set and an equal number of data samples per class. In contrast, the local data distributions in non-i.i.d. settings are more complex, with varying local training set sizes and imbalanced data samples per class. Following the settings in Xu et al. (2022), we create a non-i.i.d scenario. We first sample a Bernoulli random variable $\Phi_{kc} \sim \text{Bernoulli}(p)$ for each client $k$ and class $c$, where $\Phi_{kc} = 1$ represents that client $k$ has class $c$ and $\Phi_{kc} = 0$ otherwise. We then distribute the data samples of class $c$ among the clients with $\Phi_{*c} = 1$ using Latent Dirichlet Allocation (LDA), assigning the partition of the class $c$ samples, where $\boldsymbol{\alpha}_c$ is a vector of length $\sum_{k=1}^{N} \Phi_{kc}$ with positive elements $\alpha_{Dir}$. To ensure that no clients are left without data, we allocate at least one sample of class $c$ to each client $k$ with $\Phi_{kc} = 1$ regardless of $\boldsymbol{q}_c$.

**Noise level**  We synthetically design scenarios with different levels of label noise among clients, following Xu et al. (2022). We first introduce the parameter $\rho$, which represents the ratio of clients with label noise in their datasets. We then assign the noise rate of each noisy client based on $\tau$, the lower bound of each client's noise rate. We sample the noise rate $r_k$ of noisy client $k$ uniformly at random from the range $\mathcal{U}(\tau, 1)$.

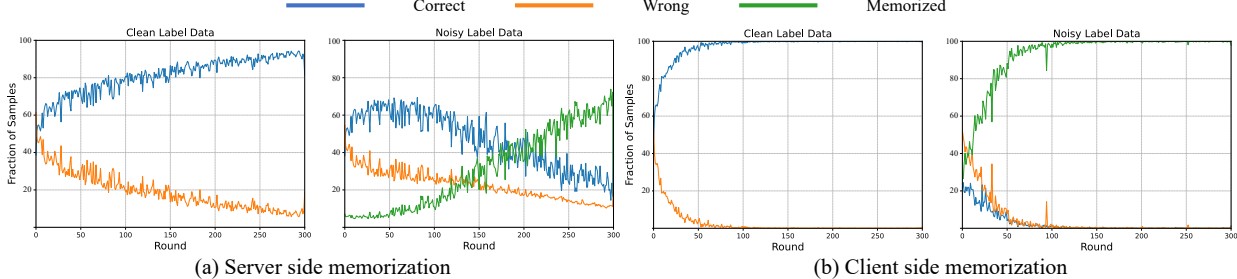

Figure 2: (a) Server side memorization with $\mathcal{L}_{CE}$, and (b) client side memorization with $\mathcal{L}_{CE}$ on CIFAR-10 of the i.i.d. setting, with symmetric noise of $(\rho, \tau) = (0.8, 0.0)$. In (b), the fraction values are calculated by averaging the values contributed solely by participating noisy clients in each round.

Finally, we randomly select $r_k n_k$ data samples and reassign their labels to create noise. It is worth noting that, while the original paper by FedCorr treated $\rho$ and $r_k$ as probabilities (through Bernoulli sampling), our implementation treats them as fixed ratios.

**Noise type** We examine two forms of label noise: symmetric and asymmetric noise (Patrini et al., 2017). For symmetric noise, we randomly and uniformly choose one class among the $C$ classes and reassign $\boldsymbol{y}_{ki}$ as an i.i.d. random one-hot vector. As for asymmetric noise, it mimics human-like errors that happen for similar classes (e.g., cat $\leftrightarrow$ dog, truck $\rightarrow$ automobile). We reassign $\boldsymbol{y}_{ki}$ based on the class of $\boldsymbol{x}_{ki}$, the initial value of $\boldsymbol{y}_{ki}$, for asymmetric noise.

## 4 Method

### 4.1 In the view of memorization in FL

In this subsection, we delve into the phenomenon of memorization in FL. Despite numerous studies in CL exploring memorization, the topic remains notably unexplored in the context of FL. To provide a clear understanding, we distinguish between two types of memorization by referring to the definition of memorization in the field of learning with noisy labels in CL (Liu et al., 2020):

- **Global memorization (server side)** transpires after federated aggregation when the accuracy of the training dataset is calculated using the server model.

- **Local memorization (client side)** takes place immediately after a local model update and refers to the accuracy computed on the training dataset using a personalized model that has been updated locally before federated aggregation occurs.

Figure 2 showcases both local and global memorization. In this context, we estimate memorization as instances where the model's predictions coincide with incorrect labels. We observe that the accuracy curves of correct samples in both clean and noisy label plots increase during the early stages. However, in the noisy label plot, the accuracy of correct samples declines after reaching a certain point, while the level of memorization rises. In contrast, the clean label curve remains stable (Figure 2 (a)). In both cases, the count of wrong-predicted samples consistently decreases. Alongside examining the server-side, we also probe into the client-side memorization. Owing to the scarce amount of data available to each client, memorization becomes more prominent, regardless of whether the data is clean or noisy (Figure 2 (b)). Specifically, the proportion of samples with incorrect labels memorized by the local model increases rapidly as the round progresses, eventually reaching 100%. This signifies that the local model has memorized all of the noisy labels present in the client's local dataset.

Based on this observation, we define memorization from both local and global perspectives, capturing severity and prevalence, respectively. Catastrophic memorization occurs at the client level when a model nearly

---

**Algorithm 1:** FLR

---

INPUT : Neural network $\mathcal{M}(\cdot)$, server model parameter $\theta_{server}$, randomly initialized parameter $\theta_0$, client $k$'s
model parameter $\theta_k$, global round $T$, local training epoch $E$, learning rate $\eta$, client $k$'s dataset
$\mathcal{D}^k = \{(\boldsymbol{x}_{k_i}, \boldsymbol{y}_{k_i})\}_{i=1}^{n_k}$, balancing hyperparameters for FLR $\alpha, \beta, \gamma$

1: $\theta_{server} \leftarrow \theta_0$      // Initialization
   /* Phase 1: Warmup FedAvg with standard CE loss */
2: **for** $t \leftarrow 0, \ldots, T-1$ **do**
3:    $S^t \leftarrow$ SAMPLECLIENTS
4:    **for** each client $k \in S^t$ in parallel **do**
5:       $\theta_k \leftarrow \theta_{server}$
6:       **for** $e \leftarrow 0, \ldots, E-1$ **do**
7:          $\boldsymbol{p}_{k_1}, \ldots, \boldsymbol{p}_{k_{n_k}} \leftarrow \mathcal{M}_{\theta_k}(\boldsymbol{x}_{k_1}), \ldots, \mathcal{M}_{\theta_k}(\boldsymbol{x}_{k_{n_k}})$
8:          $\theta_k \leftarrow \theta_k - \eta \nabla(-\frac{1}{n_k} \sum_{i=1}^{n_k} \sum_{c=1}^{C} \boldsymbol{y}_{k_i}^{(c)} \log \boldsymbol{p}_{k_i}^{(c)}))$
9:       **end for**      // Client-Update with standard CE loss
10:    **end for**
11:    $\theta_{server} \leftarrow \sum_{k \in S^t} \frac{|\mathcal{D}^k|}{\sum_{k' \in S^t} |\mathcal{D}^{k'}|} \theta_k$      // Aggregation
12: **end for**
   /* Phase 2: FedAvg with FLR loss*/
13: **for** $t \leftarrow 0, \ldots, T-1$ **do**
14:    $S^t \leftarrow$ SAMPLECLIENTS
15:    **for** each client $k \in S^t$ in parallel **do**
16:       $\theta_k \leftarrow \theta_{server}$
17:       $\boldsymbol{p}_{k_1}^{server}, \ldots, \boldsymbol{p}_{k_{n_k}}^{server} \leftarrow \mathcal{M}_{\theta_{server}}(\boldsymbol{x}_{k_1}), \ldots, \mathcal{M}_{\theta_{server}}(\boldsymbol{x}_{k_{n_k}})$      // $\theta_{server}$ is frozen
18:       **for** $e \leftarrow 0, \ldots, E-1$ **do**
19:          $\boldsymbol{p}_{k_1}, \ldots, \boldsymbol{p}_{k_{n_k}} \leftarrow \mathcal{M}_{\theta_k}(\boldsymbol{x}_{k_1}), \ldots, \mathcal{M}_{\theta_k}(\boldsymbol{x}_{k_{n_k}})$
20:          **for** $i \leftarrow 1, \ldots, n_k$ **do**
21:             $\boldsymbol{s}_{k_i} \leftarrow \beta \boldsymbol{s}_{k_i} + (1-\beta) \boldsymbol{p}_{k_i}^{server}$      // Global running average mean
22:             $\boldsymbol{m}_{k_i} \leftarrow \gamma \boldsymbol{m}_{k_i} + (1-\gamma) \boldsymbol{p}_{k_i}$      // Local running average mean
23:             $\boldsymbol{t}_{k_i} \leftarrow \alpha \boldsymbol{s}_{k_i} + (1-\alpha) \boldsymbol{m}_{k_i}$      // Mixture
24:          **end for**
25:          $\theta_k \leftarrow \theta_k - \eta \nabla(-\frac{1}{n_k} \sum_{i=1}^{n_k} \sum_{c=1}^{C} \boldsymbol{y}_{k_i}^{(c)} \log \boldsymbol{p}_{k_i}^{(c)} + \frac{\lambda}{n_k} \sum_{i=1}^{n_k} \log(1 - \langle \boldsymbol{p}_{k_i} \cdot \boldsymbol{t}_{k_i} \rangle))$
26:       **end for**      // Client-Update with FLR
27:    **end for**
28:    $\theta_{server} \leftarrow \sum_{k \in S^t} \frac{|\mathcal{D}^k|}{\sum_{k' \in S^t} |\mathcal{D}^{k'}|} \theta_k$      // Aggregation
29: **end for**

---

memorizes its noisy training data, leading to suboptimal generalization and overfitting to incorrect labels. Conversely, common memorization refers to shared memorization across clients from the global perspective, emerging when numerous client models exhibit similar patterns of memorizing noisy labels, indicating a more pervasive issue affecting the overall FL process.

### 4.2 Global & local moving average label mixture

To combat these memorization issues, we introduce a regularization term that penalizes the local prediction based on the difference between the model's output and synthetic pseudo labels:

$$\mathcal{L}_{FLR}^k(\theta) := \mathcal{L}_{CE}^k(\theta) + \frac{\lambda}{n_k} \sum_{i=1}^{n_k} \log(1 - \langle \boldsymbol{p}_{k_i} \cdot \boldsymbol{t}_{k_i} \rangle)$$

$$\mathcal{L}_{CE}^k(\theta) := -\frac{1}{n_k} \sum_{i=1}^{n_k} \sum_{c=1}^{C} \boldsymbol{y}_{k_i}^{(c)} \log \boldsymbol{p}_{k_i}^{(c)}$$

(1)

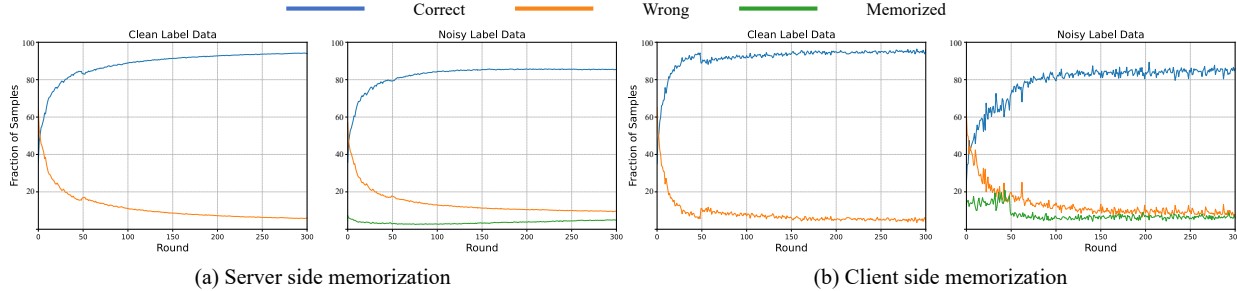

Figure 3: (a) Server-side memorization with $\mathcal{L}_{FLR}$, and (b) client-side memorization with $\mathcal{L}_{FLR}$ under the same setting used in Figure 2.

$$\boldsymbol{s}_{k_i} \leftarrow \beta \boldsymbol{s}_{k_i} + (1-\beta)\boldsymbol{p}_{k_i}^{server} \quad \triangleright \text{Global}$$
$$\boldsymbol{m}_{k_i} \leftarrow \gamma \boldsymbol{m}_{k_i} + (1-\gamma)\boldsymbol{p}_{k_i} \quad \triangleright \text{Local}$$
$$\boldsymbol{t}_{k_i} \leftarrow \alpha \boldsymbol{s}_{k_i} + (1-\alpha)\boldsymbol{m}_{k_i} \quad \triangleright \text{Mixture}$$

where $\theta$ denotes the parameters of the neural network, $\boldsymbol{p}_{k_i}$ is the local prediction, $\boldsymbol{t}_{k_i}$ is the pseudo label vector for a client $k$'s $i$th input, $\boldsymbol{s}_{k_i}$ is the global running average prediction, $\boldsymbol{p}_{k_i}^{server}$ is the global model's prediction, $\boldsymbol{m}_{k_i}$ is a local running average prediction, and $c$ is the index of the vector, which satisfies with $1 \leq c \leq C$. The logarithm component serves as a regularization term that aligns $\boldsymbol{p}$ in the same direction as $\boldsymbol{t}_{k_i}$. Synthetic pseudo labels $\boldsymbol{t}_{k_i}$ are generated by merging a local running average prediction with the server's running average prediction, called Federated Label-mixture Regularization (FLR). For an initial prediction of a client $k$'s $i$th input, $\boldsymbol{s}_{k_i}$ and $\boldsymbol{m}_{k_i}$ are set to be equal to $\boldsymbol{p}_{k_i}^{server}$ and $\boldsymbol{p}_{k_i}$, respectively. Since the server receives randomly augmented data during each local iteration, $\boldsymbol{p}_{k_i}^{server}$ undergoes slight changes. The number of updates to $\boldsymbol{t}_{k_i}$ depends on the number of clients participating in each round. If a client does not participate, the most recent version of $\boldsymbol{t}_{k_i}$ is stored locally, and this value is used for updates when the client is selected from the server. Algorithm 1 illustrates the overall procedure within the general FL framework.

The parameters $\alpha$, $\beta$, and $\gamma$ in Eq. (1) play critical roles in balancing the influence of global and local information. Here, $\alpha$ controls the weighting between the global and local running averages, influencing the model's bias towards more generalized or personalized predictions. A higher $\alpha$ value emphasizes the global context, reducing overfitting to noisy local data but possibly underfitting local nuances. Conversely, $\beta$ and $\gamma$ manage the decay rates of the global and local running averages, respectively. Specifically, $\beta$ adjusts how quickly the global information is integrated into the model updates, while $\gamma$ dictates the retention of local data characteristics in the learning process. These hyperparameters are pivotal in optimizing the trade-off between memorization and generalization, ensuring that the model remains robust to label noise and data heterogeneity across different clients. Their values should be chosen based on empirical evaluations and the specific characteristics of the dataset, as thoroughly discussed in Subsection 4.3, which examines their impacts through a series of ablation studies.

**Tackling confirmation bias** The phenomenon of models overfitting to noisy labels, known as memorization, can be viewed as a form of confirmation bias, where incorrect pseudo-labels predicted by the network are erroneously reinforced (Tarvainen & Valpola, 2017; Arazo et al., 2020). FLR uses pseudo labels $\boldsymbol{t}_{k_i}$ —derived from a blend of local and global running average predictions—to temper this bias. The local predictions capture client-specific noise patterns, while the global predictions reflect the broader data distribution, together creating a comprehensive and resilient learning signal. The balancing parameter $\alpha$ plays a pivotal role, meticulously adjusting the blend of local and global perspectives to ensure optimal noise resistance and overall performance. While local running average prediction alone can be effective in centralized settings, it may exacerbate local overfitting more swiftly in FL contexts due to data heterogeneity and data shortage. Our strategy of blending local and global predictions can thus be particularly advantageous in such situations. More details are in Subsection 4.3.

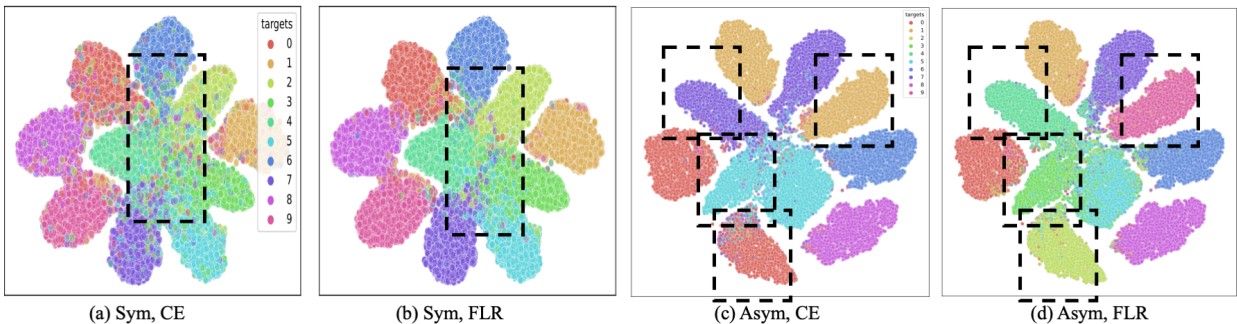

| (a) Sym, CE | (b) Sym, FLR | (c) Asym, CE | (d) Asym, FLR |

Figure 4: t-SNE mapping view of the penultimate layer's feature outputs. Colors represent the noisy label $y$ and the ground truth label $y^*$.

Table 1: Server side memorization and client side memorization at last round on CIFAR-10 of non-i.i.d. setting ($\alpha_{dir} = 1.0$) with symmetric noise $(\rho, \tau) = (1.0, 0.0)$ and i.i.d. setting with symmetric noise $(\rho, \tau) = (0.8, 0.0)$. Vanilla denotes no use of the regularization in Eq. (1).

| Heterogeneity | Pseudo Label | Global memorization (server-side) | | | | | Local memorization (client-side) | | | | | Test Acc. |
|---|---|---|---|---|---|---|---|---|---|---|---|---|
| | | Clean label | | Noisy label | | | Clean label | | Noisy label | | | |
| | | Correct | Wrong | Correct | Wrong | Memorized | Correct | Wrong | Correct | Wrong | Memorized | |
| non-i.i.d. | Vanilla | 69.10 | 30.90 | 32.96 | 34.61 | 32.43 | 95.38 | 4.62 | 12.40 | 7.10 | 80.50 | 66.54 |
| | Local Only | 80.11 | 19.89 | 62.48 | 26.52 | 11.00 | 95.61 | 4.39 | 40.19 | 10.04 | 49.77 | 81.12 |
| | Global Only | 90.67 | 9.32 | 74.63 | 11.31 | 14.05 | 94.79 | 5.21 | 72.56 | 6.72 | 20.72 | 82.09 |
| | Mixture (FLR) | 88.04 | 11.96 | 70.36 | 21.51 | 8.13 | 97.64 | 2.36 | 71.58 | 13.03 | 15.39 | **83.04** |
| i.i.d. | Vanilla | 90.92 | 9.08 | 16.57 | 11.16 | 72.26 | 99.96 | 0.04 | 0.00 | 0.00 | 100.00 | 67.87 |
| | Local Only | 94.88 | 5.12 | 80.34 | 5.46 | 14.19 | 99.55 | 0.45 | 21.22 | 0.88 | 77.90 | 83.88 |
| | Global Only | 96.38 | 3.62 | 77.93 | 7.56 | 14.51 | 99.27 | 0.73 | 58.41 | 2.41 | 39.17 | 85.82 |
| | Mixture (FLR) | 95.70 | 4.30 | 79.30 | 9.35 | 11.35 | 98.24 | 1.76 | 81.96 | 2.07 | 15.97 | **87.07** |

**Dissecting gradients** The regularization term's logarithm in Eq. (1) works to balance out the exponential nature of the softmax function that is present in the probability of $p_{k_i}$. Similar to Liu et al. (2020), the gradient of our method is calculated as follows:

$$\nabla \mathcal{L}_{FLR}(\theta) = \frac{1}{n} \sum_{i=1}^{n} \nabla \mathcal{N}_{\boldsymbol{x}_{k_i}} ( \overbrace{\boldsymbol{p}_{k_i} - \boldsymbol{y}_{k_i}}^{\text{gradient of } \mathcal{L}_{CE}} + \overbrace{\lambda \boldsymbol{g}_{k_i}}^{\text{gradient of the regularization}} )$$

$$\text{where } \boldsymbol{g}_{k_i}^{(c)} := \frac{\boldsymbol{p}_{k_i}^{(c)}}{1 - \langle \boldsymbol{p}_{k_i}, \boldsymbol{t}_{k_i} \rangle} \sum_{r=1}^{C} (\boldsymbol{t}_{k_i}^{(r)} - \boldsymbol{t}_{k_i}^{(c)}) \boldsymbol{p}_{k_i}^{(r)}$$

(2)

where $\nabla \mathcal{N}_{\boldsymbol{x}_{k_i}}$ is the Jacobian matrix of the neural network $\mathcal{N}$ having parameters $\theta$ for a client $k$'s $i$th input with respect to the $\theta$. Suppose the actual class is $c^*$. In the early training phase, the $c^*$th entry of $\boldsymbol{s}_{k_i}$ is dominant. Setting $\alpha \geq 0.5$ results in a negative $c^*$th entry of $\boldsymbol{g}_{k_i}$, since the server memorizes slower than clients, preventing harmful memorization by noisy clients. Moreover, in the case of clean labels, the gradient of $\mathcal{L}_{CE}$ term $\boldsymbol{p}_{k_i} - \boldsymbol{y}_{k_i}$ decreases quickly due to small dataset size, potentially leading to dominance by wrong labels in the gradient. The term $\boldsymbol{g}_{k_i}$ mitigates this by maintaining the magnitude of coefficients on clean label examples. For mislabeled examples, the gradient of $\mathcal{L}_{CE}$ term $\boldsymbol{p}_{k_i}^{(r)} - \boldsymbol{y}_{k_i}^{(r)}$ is positive, and adding the negative term $\boldsymbol{g}_{k_i}^{(r)}$ reduces the coefficients on these examples, minimizing their gradient impact. Beyond a convergence point, the server might not curtail local memorization due to overfitting. The term $\boldsymbol{m}_{k_i}$ can be added at this stage to counteract local memorization with local running average predictions.

**Warmup FedAvg (phase1)** Our analysis of Figure 2 (a) reveals that in the initial rounds, there is no significant server-level memorization. This observation justifies the application of a warm-up phase in our

methodology, aligning with the standard practices in previous LNL and FNL research (Jiang et al., 2022; Xu et al., 2022; Kim et al., 2022; Liu et al., 2020; Li et al., 2020a).

**Experimental results**   Our empirical observations indicate that FLR effectively reduces both local and global memorization when compared to models trained using the general FedAvg method, which relies solely on cross-entropy loss. As demonstrated in Figure 3 (a), the learning curve is smoother for FLR, and the amount of memorization is considerably lower. In addition, the model trained with FLR accurately identifies the ground truth labels of noisy samples as the rounds progress, while maintaining the memorized ratio of incorrect labels below 10% (Figure 3 (b)). We employ warm-up training for regularization with server prediction during the first 50 epochs and switch to local running average prediction regularization thereafter. This transition causes the inflection point in the graph to appear shortly after warm-up completion in both Figure 3 (a) and (b). Although the accuracy for clean data experiences a slight decrease, the memorization of noisy data is significantly reduced.

**Analysis according to data heterogeneity**   Table 1 presents the server-side and client-side memorization for both i.i.d. and non-i.i.d. settings when using local and global moving averages. In i.i.d. settings, employing local moving averages alone effectively mitigates global memorization but not local memorization. In contrast, in non-i.i.d. scenarios, using the global moving average has a more significant impact on reducing memorization. This can be attributed to the fact that when clients have similar data distributions, local information is sufficient to prevent memorization in the global model. However, in non-i.i.d. situations, relying solely on local representation is inadequate for preventing global model memorization, making the use of the global moving average more advantageous. In both cases, utilizing a mixture of local and global moving average predictions as synthetic pseudo labels produced the best results, as shown in the table's Mixture rows. Although the degree of data heterogeneity may affect the outcome, a balanced mixture of global and local predictions seems to be more effective than adopting biased pseudo labels from either side exclusively.

**Qualitative analysis**   Figure 4 provides t-SNE mappings of the penultimate layer's outputs, where each dot is visualized according to the predictions by the model trained with standard CE loss (Figure 4 (a) and (c)) and the model trained with FLR (Figure 4 (b) and (d)). The color corresponds to the ground truth labels. This visual analysis reveals that FLR significantly enhances cluster fidelity to reflect true class information of the samples. Figure 4 (a) and (c) depict the clustering under the CE loss, where color discrepancies indicate noisy label misclassifications. In contrast, Figure 4 (b) and (d) illustrate substantial improvements under FLR; clusters are well-separated according to their true classes. Black dashed boxes highlight the visible transformations from the CE to the FLR model, illustrating the robustness of FLR in correcting misclassifications. For instance, the transition from scattered and inconsistently colored clusters in Figure 4 (a) to more cohesive and correctly labeled groups in Figure 4 (b) not only evidences a decrease in label noise but also reflects substantial qualitative enhancements in model performance. The t-SNE plots in Figure 4 provide an intuition regarding how FLR successfully addresses label noise.

## 4.3 Implementation

Figure 1 provides an illustration of how our method affects model updates. It is widely recognized that using a running average (i.e., temporal ensembling (Laine & Aila, 2017)) is more effective than relying on static model output. As a result, the baseline version of our method follows Eq. (1). We empirically investigate the effects of the components $\alpha, \beta, \gamma$ (Figure 5). Insights for FLR's effective label noise handling in FL settings include:

- **Influence of $\alpha$ and $\gamma$:** We observe that higher values of $\alpha$ consistently yield better results, indicating a robust influence of global model predictions on local updates (Figure 5 (a)). An $\alpha$ setting of 0.9 acts as a potent regularization mechanism, akin to knowledge distillation, guiding local models with the refined, effective knowledge from the global model, thus bolstering system robustness as illustrated in Figure 2. At an $\alpha$ value of 1, the model fully embraces global predictions, epitomizing a comprehensive knowledge distillation strategy. This provides profound insights into the extent of global model influence on local training dynamics. Meanwhile, a $\gamma$ setting between 0.5 and 0.7

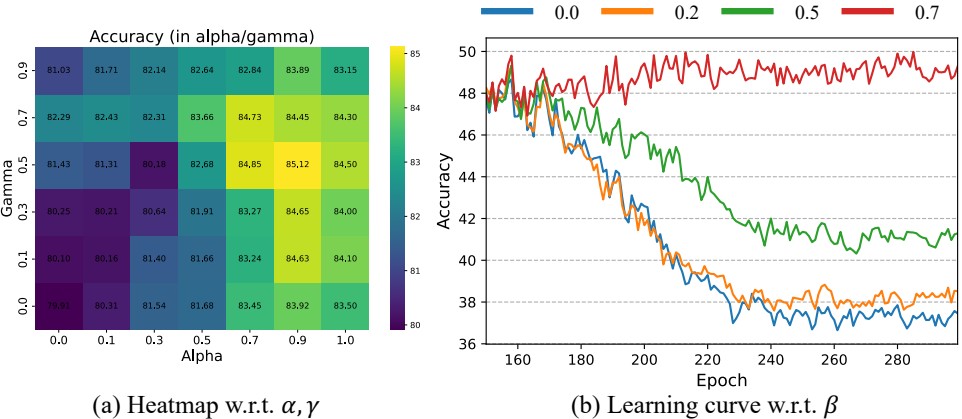

(a) Heatmap w.r.t. $\alpha, \gamma$             (b) Learning curve w.r.t. $\beta$

Figure 5: (a) Heatmaps with respect to $\alpha, \gamma$ at $(\rho, \tau) = (0.6, 0.5)$ on CIFAR-10, (b) learning curves according to the changes of $\beta$ at $(\rho, \tau) = (1.0, 0.5)$ on CIFAR-10.

fine-tunes the update frequency, striking a balance between historical accuracy and mitigation of local noise. This measured approach parallels careful learning rate adjustments, promoting gradual adaptation of local models to global insights and preventing destabilizing shifts.

- **Preventing global memorization with** $\beta$**:** In the presence of extreme noise, global memorization tends to occur in the latter half of the training process. It is apparent that higher $\beta$ settings result in improved performance in highly noisy environments. This trend highlights the efficacy of increasing $\beta$ to mitigate global memorization risks by incorporating the moving average predictions from the global model (Figure 5 (b)*).

**Implications of** $\alpha, \beta, \gamma$    We briefly discuss the implications of our methods:

- $\alpha$: Balances between local and global memorization. A lower $\alpha$ leads the model to focus more on local memorization, capturing specific patterns unique to each client, but risks overfitting. A higher $\alpha$ shifts the emphasis towards global memorization, encouraging learning of universal patterns across the dataset.

- $\beta$: Addresses global memorization in the server model, especially after several epochs, by employing temporal ensembling. This helps the model to maintain an overarching perspective on the learned patterns.

- $\gamma$: Aims to control local memorization. While the server model counters local memorization initially, $\gamma$ becomes essential when local memorization re-emerges at a certain convergence point. However, applying $\gamma$ too early can interfere with the learning trajectory, thus it is strategically implemented after reaching a certain level of learning.

To effectively tackle memorization, our approach dynamically adjusts the influence of parameters $\boldsymbol{m}_{k_i}$ and $\boldsymbol{s}_{k_i}$ throughout training. Initially, $\boldsymbol{m}_{k_i}$ is more influential to support early learning stages, but as training progresses, $\boldsymbol{s}_{k_i}$ takes precedence for temporal ensembling to address memorization challenges. This dynamic is managed through a sophisticated scheduler:

$$\alpha(r) = \alpha \times \frac{r}{R}, \quad \beta(r) = \begin{cases} 0 & \text{if } r < \frac{R}{2} \\ \beta & \text{else} \end{cases}, \quad \gamma(r) = \begin{cases} 0 & \text{if } r < R_w \\ \gamma & \text{else} \end{cases}$$

---

*The selection of $\gamma$ values from the 150th epoch in Figure 5 (b) is informed by empirical evidence showing a shift in local memorization patterns around this stage across benchmarks. This period marks the initiation of our method's application, a decision based on its proven efficacy in enhancing model performance. This strategic timing enables precise monitoring and reporting on the method's impact during critical learning phases, where interventions are most crucial.

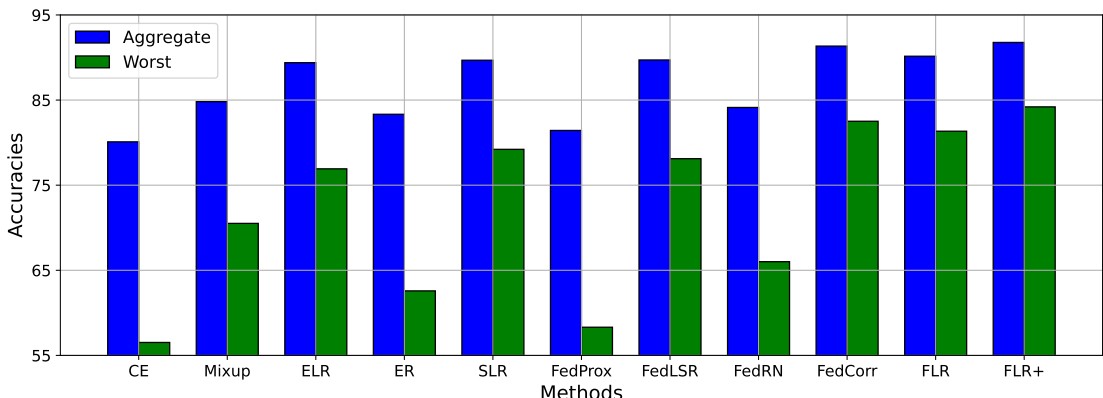

Figure 6: Test accuracies for each method, evaluated on CIFAR10N dataset (Wei et al., 2021). ELR is implemented with FedAvg. 'Aggregate' and 'Worst' denote different noise type of CIFAR 10N dataset.

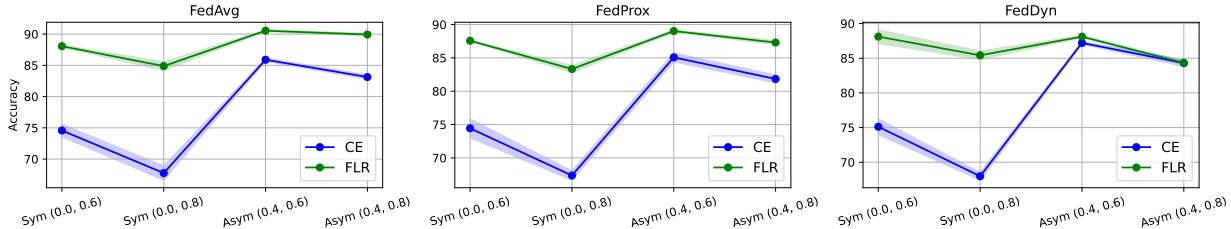

Figure 7: Collaborations with server-aware FL methods (FedProx (Li et al., 2020c) and FedDyn (Durmus et al., 2021)) on CIFAR-10 with i.i.d. setting.

Here, $R$ denotes the total communication rounds, $r$ is the current round, and $R_w$ is the warm-up period for $\beta$. We set $(\lambda, \alpha, \beta, \gamma)$ typically to $(2.0, 0.9, 0.7, 0.5)$ with $R_w$ at 50, with further details provided in the Appendix.

The linear scheduler for $\alpha$ delicately manages the balance between local and global learning. Initially, it enhances local learning to compensate for the global model's lower accuracy but gradually shifts to support global learning as the model's noise handling improves. $\beta$ and $\gamma$ start at zero to accommodate initial learning stages and gradually increase to harmonize the integration of local and global predictions as training advances.

For hyperparameter selection crucial to the effective deployment of our methodology, we recommend tailoring these parameters based on specific data characteristics and learning objectives. Comprehensive guidance on selecting and optimizing these parameters, along with practical application tips, are detailed in the Appendix.

## 5 Experiments

### 5.1 Experimental setup

We evaluate our methods on two standard benchmark datasets, CIFAR-10 and CIFAR-100 (Krizhevsky et al., 2009), and two real-world datasets, CIFAR10N (Wei et al., 2021) and Clothing1M (Xiao et al., 2015), with varying numbers of clients. We conduct experiments with 100 clients for CIFAR-10 and CIFAR10N, 50 clients for CIFAR-100, and 500 clients for Clothing1M. We consider both i.i.d. (CIFAR-10/100) and non-i.i.d. settings (CIFAR-10/100, CIFAR10N, Clothing1M); non-iidness is parameterized by $p$ and $\alpha_{Dir}$. As CIFAR-10/100 do not inherently contain label noise, we introduce synthetic label noise, where the level of noise is parameterized by $\rho$ and $\tau$. CIFAR10N and Clothing1M naturally contain label noise, with data samples randomly distributed among clients. $p$, $\alpha_{Dir}$, $\rho$, and $\tau$ are elaborated upon in Section 3. To contrast between CL and FL, we also compare our methods with conventional LNL methods in centralized settings,

Table 2: Average and standard deviation of the best test accuracy of 5 trials for each method, evaluated on Clothing1M dataset (Xiao et al., 2015) in federated scenarios with uniformly random distributed settings. FLR$^+$ denotes the accuracy of the FLR with FedCorr framework.

| FedAvg | GCE | Co-teaching | ELR | FedProx | RoFL | FedLSR | FedRN | FedCorr | FLR | FLR$^+$ |
|---|---|---|---|---|---|---|---|---|---|---|
| 67.73±0.18 | 63.57±2.85 | 67.67±0.10 | 68.68±0.16 | 66.99±0.13 | 67.68±0.17 | 69.38±0.15 | 68.54±0.16 | 69.93±0.11 | 69.76±0.19 | **70.11±0.14** |

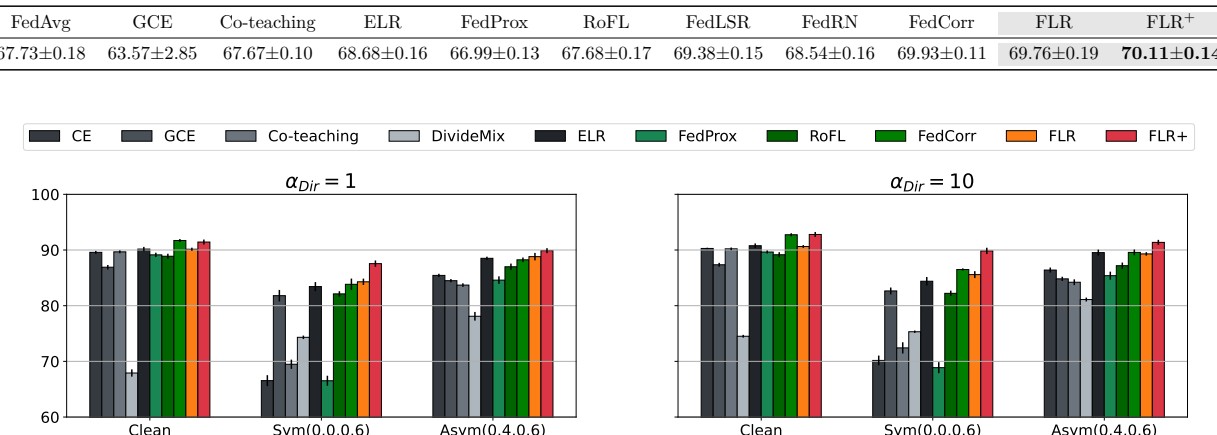

Figure 8: Average of the best test accuracy of 5 trials for each method, on CIFAR-10 with non-i.i.d. setting by varying the concentration parameter $\alpha_{Dir}$ under different noise levels. The x-axis indicates the noise level by changing the $(\tau, \rho)$.

implemented following Xu et al. (2022). In the centralized scenario, we employ a dataset that has been corrupted using the exact same scheme as in the federated scenario. Further implementation specifics are available in the Appendix.

## 5.2 Results on real world noisy datasets

**CIFAR10N** Our experiments demonstrate that FLR surpasses traditional LNL methods utilized in FL. Notably, FLR's effectiveness is further amplified when combined with FedCorr, indicating its robustness and adaptability across diverse FL settings. Figure 6 showcases FLR's significant advantage over other label regularization techniques.

**Clothing1M** FLR consistently outperforms other LNL methods. Although the accuracy differences among the methods are modest, FLR's performance stands out. This consistency in performance is particularly noteworthy considering the use of a pre-trained model for the initial global model in these experiments. The results, detailed in Table 2, underscore FLR's capability to improve model accuracy in FL settings, even under conditions with pretrained models.

## 5.3 Results on CIFAR datasets

### 5.3.1 I.I.D. setting

Our comprehensive evaluation on CIFAR-100 under an i.i.d. setting (Table 3) showcases FLR's remarkable performance superiority across diverse label noise settings, including both symmetric and asymmetric noise scenarios. We observe that even well-established CL methods like DivideMix (Li et al., 2020a), while robust in CL, do not seamlessly translate their effectiveness into the federated context. This finding resonates with prior literature on FL (FedCorr (Xu et al., 2022), FedRN (Kim et al., 2022)). Notably, Generalized Cross Entropy (GCE) (Zhang & Sabuncu, 2018) exhibits strong performance, particularly in high symmetric noise scenarios, yet FLR consistently outperforms it in most cases. Combining FLR with FedCorr (i.e., FLR$^+$) yields a synergistic effect, leading to the highest test accuracies across almost all noise settings, except for the noise-free case. This combination demonstrates FLR's versatility and potential for enhancing FL frameworks under diverse noise conditions.

Table 3: Average and standard deviation of the best test accuracy of 5 trials for each method, on CIFAR-100 with i.i.d. setting and different noise levels.

| Setting | Method | Symmetric | | | | Asymmetric |
|---|---|---|---|---|---|---|
| | | $\tau = 0.0$ | | $\tau = 0.5$ | | $\tau = 0.4$ |
| | | $\rho = 0.6$ | $\rho = 0.8$ | $\rho = 0.6$ | $\rho = 0.8$ | $\rho = 0.6$ |
| Centralized | CE | 50.62±1.42 | 46.30±1.88 | 43.02±1.08 | 32.92±1.26 | 55.74±0.12 |
| | GCE | 61.69±0.77 | 59.06±0.92 | 57.25±0.58 | 48.91±1.32 | 62.48±0.51 |
| | Co-teaching | 42.68±1.24 | 38.45±1.88 | 36.38±0.55 | 27.18±0.81 | 41.90±1.33 |
| | DivideMix | **69.17±0.83** | **65.76±1.24** | **64.91±0.81** | **54.94±1.02** | **69.95±0.50** |
| | ELR | 61.26±1.59 | 56.58±2.09 | 53.09±1.21 | 39.71±1.10 | 69.57±0.16 |
| Federated | FedAvg | 42.99±1.29 | 36.68±2.02 | 34.53±1.12 | 25.48±0.80 | 51.79±0.32 |
| | GCE | 48.51±1.37 | 43.18±1.54 | 40.55±1.65 | 30.96±1.14 | 54.60±0.48 |
| | Co-teaching | 44.74±1.23 | 38.81±1.75 | 36.55±0.83 | 27.61±0.97 | 53.30±0.82 |
| | DivideMix | 55.10±1.20 | 52.23±0.61 | 48.88±1.34 | 43.57±0.81 | 55.50±0.29 |
| | ELR | 51.02±1.58 | 43.47±2.00 | 40.58±1.71 | 28.50±1.03 | 56.98±0.67 |
| | FedProx | 40.61±1.47 | 34.36±1.95 | 32.71±0.99 | 24.04±0.85 | 49.11±0.68 |
| | RoFL | 43.44±1.36 | 36.61±2.12 | 38.61±1.13 | 26.53±1.07 | 51.83±0.65 |
| | FedLSR | 60.51±1.21 | 45.62±1.88 | 41.51±1.90 | 30.35±0.84 | 58.95±0.92 |
| | FedRN | 43.10±0.90 | 36.58±2.34 | 38.46±1.88 | 24.32±1.00 | 53.04±0.40 |
| | FedCorr | 53.23±1.81 | 46.25±1.95 | 44.13±0.61 | 31.43±1.54 | 59.61±0.56 |
| | FLR | 59.13±1.61 | 54.18±1.87 | 51.12±1.46 | 40.15±1.34 | 65.21±0.56 |
| | FLR$^+$ | **67.85±0.91** | **64.12±2.01** | **64.64±0.38** | **54.10±1.01** | **69.52±0.71** |

**Collaborations with server-aware FL methods**   We also delve into how FLR collaborates with server-aware FL methods. Our findings, presented in Figure 7, demonstrate that integrating FLR with these methods significantly boosts accuracy in various noisy client scenarios. This enhancement is observed consistently across different ratios of noisy clients, highlighting FLR's capability to complement server-aware strategies. This synergy is particularly crucial in federated environments where client data is diverse and often unreliable, showcasing FLR's versatility and effectiveness in improving outcomes in complex FL settings.

### 5.3.2   Non-I.I.D. setting

**Concentration parameter in Dirichlet distribution** $\alpha_{Dir}$   Figure 8 shows the results by changing the data heterogeneity $\alpha_{Dir}$ at $p = 1.0$. Overall, it can be seen that the performance is lower when $\alpha_{Dir} = 1$ (local data heterogeneity ↑) than when $\alpha_{Dir} = 10$ (local data heterogeneity ↓). It is confirmed that FLR consistently shows better performance at most noise levels and there seems more a clear difference, especially when data heterogeneity becomes more severe.

## 6   Conclusion

This paper presents Federated Label-Mixture Regularization (FLR), a simple but effective method designed to tackle label noise in FL environments. Our work is grounded in two key contributions: a detailed exploration of memorization dynamics in FL, distinguishing between server-side and client-side memorization, and the development of FLR to address these specific challenges. By creating novel pseudo labels, which are a combination of global running average predictions and local running average predictions, FLR effectively mitigates these memorization issues. Through comprehensive experiments, FLR demonstrates a marked improvement in global model accuracy across both i.i.d. and non-i.i.d. data settings. While it shows promising results, we acknowledge that FLR may not be optimally suited for all types of label noise. Future research could delve into exploring complementary regularization techniques to address various noise structures. The potential impact of FLR on real-world applications, especially in sectors like healthcare and financial services where data privacy is paramount, is significant.

**Acknowledgments**

This work was partly supported by Institute for Information & communications Technology Promotion (IITP) grant funded by the Korea government (MSIT) [No.RS-2019-II190075, Artificial Intelligence Graduate School Program (KAIST), 10%, and No. 2022-0-00871, Development of AI Autonomy and Knowledge Enhancement for AI Agent Collaboration, 90%].

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

## A    Overview of Appendix

We present additional details, results, and experiments that are not included in the main paper.

## B    Ethics Statement

As we advance the field of Federated Learning (FL) with our novel Federated Label-mixture Regularization (FLR) method, it is crucial to consider its ethical implications. We address potential concerns in the areas of privacy, fairness, environmental impact, and potential misuse.

**Privacy and data security**   FLR, like other FL methods, is designed to maintain the privacy of clients' data by performing computations locally and only sharing model updates. This approach inherently mitigates privacy concerns arising from the transmission of raw data. However, potential risks persist with the possibility of model inversion attacks or other adversarial actions. Future work must continue to focus on strengthening the robustness of FL methods against such threats.

**Fairness**   FLR could potentially exacerbate or mitigate existing fairness issues in FL depending on how it is deployed. On one hand, if FLR is used predominantly with data from certain demographics, it may introduce bias into model predictions. On the other hand, FLR's ability to handle noisy labels may allow for the inclusion of previously excluded data, promoting a more diverse and fair model.

**Environmental impact**   FLR, like other FL methods, reduces the need for centralized data storage and computation, potentially lowering the carbon footprint associated with these processes. However, the energy consumption of local computation and communication for model updates must be carefully managed to ensure environmental sustainability.

**Potential misuses**   While FLR is designed to improve the robustness of FL methods against noisy labels, it could potentially be misused to intentionally introduce bias or misinformation into models. For instance, malicious actors could leverage our method to facilitate their objectives by manipulating the noise in labels. Hence, it is crucial to develop safeguards against such misuse.

In conclusion, while FLR contributes positively to the FL field by offering a solution to handle noisy labels, it is essential to apply it thoughtfully, considering its potential ethical implications. We encourage ongoing dialogue and scrutiny to ensure that its deployment aligns with principles of privacy, fairness, and social responsibility.

## C    Limitations

While our FLR method presents several noteworthy advancements in handling noisy labels in FL, it is crucial to acknowledge the limitations that accompany it:

1. **Performance with highly imbalanced data:** Although our FLR method demonstrates robustness in diverse data heterogeneity scenarios, its efficacy in situations with severe data imbalance across clients is an area that requires further investigation. Data imbalance here refers to the uneven distribution of data classes across different clients. For example, one client might have a majority of samples from a particular class while another client might have a very few samples of that class. This severe skewness in data distribution can lead to biased learning, where the model performs well on classes with more samples but poorly on classes with fewer samples. Although FLR has mechanisms to mitigate the effects of data heterogeneity, it is not explicitly designed to handle extreme data imbalance.

2. **Computational overhead:** While the FLR method is effective in improving the model's robustness against noisy labels, it also introduces additional computational overhead. This is primarily due to the calculation of local and global moving averages, which are central to the FLR method. The

moving average calculations require additional computation and memory resources, which might not be suitable for resource-constrained devices often participating in FL. This could potentially limit the scalability and applicability of our method in real-world FL scenarios, particularly where computational resources are a constraint. Therefore, optimizing the computational efficiency of FLR without compromising its effectiveness against noisy labels is an important direction for future research.

3. **Noise type sensitivity:** FLR has been specifically developed to tackle the challenge of label noise in FL. This label noise can occur due to various reasons, such as human errors during data labeling or miscommunication during data transfer. However, in real-world scenarios, there are other types of noise that can equally affect the learning process. For instance, feature noise refers to inaccuracies or inconsistencies in the input data, which might be caused by faulty sensors or measurement errors. Model noise, on the other hand, is related to the discrepancies between the true data generating process and the model assumptions. Since our FLR method primarily focuses on label noise, it might not exhibit the same level of efficiency when dealing with feature or model noise. Future iterations of FLR could explore these areas to provide a more comprehensive solution to noise handling in FL.

## D  Future directions

Despite the aforementioned limitations, the FLR method opens up several promising avenues for future research:

1. **Handling other noise types:** Future work can look into extending the FLR method to handle other types of noise efficiently. This would make it a more comprehensive solution for real-world federated scenarios where different types of noise coexist.

2. **Adaptation for imbalanced data:** Investigating and enhancing the performance of FLR with highly imbalanced data distribution would be a valuable future direction. Techniques like adaptive resampling or cost-sensitive learning could be integrated with our method to tackle this challenge.

3. **Applications on natural language processing:** Despite the emergent ability of Large Language Models (LLM), they also struggle with the privacy issues. Employing LLM training in federated scenario may meet noisy labels and texts with high probability, and thus our work can be a good solution in this area.

4. **Optimization of computational efficiency:** Future research could also focus on optimizing the computational efficiency of the FLR method. Reducing the computational overhead without compromising the robustness against noisy labels would make our method more practical for real-world FL scenarios.

5. **Robustness against adversarial attacks:** As the FL domain evolves, adversarial attacks pose an increasing threat to model robustness. Future work could explore how to bolster the FLR method (and FL methods, in general) to ensure robustness against adversarial attacks.

6. **Domain generalization:** Another major challenge of data heterogeneity in real-world FL application is the divergence of domain data, also known as *domain shift* (Bartholet et al., 2024). Future work should also consider the robustness to domain shift adding to the noisy labeled data.

By addressing these limitations and exploring these future directions, we can continuously refine and evolve the FLR method to better serve the ever-growing demands of FL.

### D.1  Adversarial clients in FL

Meanwhile, there exist some studies (Fu et al., 2019; Tolpegin et al., 2020; Wan & Chen, 2021; Park et al., 2021) to combat adversarial attacks on federated systems. Among them, data poisoning attack might seem

similar to FNL; few malicious clients aim to break the global model by sending local models trained with mislabeled data. However, FNL is distinct from it since every client can have some degree of label noises in FNL. From this perspective, we do not consider such works in our study.

## E    Experimental settings

The common settings are used in this paper for all baseline experiments on each datasets (Table 4).

Table 4: Datasets and the common settings. † indicates the pretrained architecture from ImageNet-1k.

| Dataset | CIFAR-10 (CIFAR-10N) | CIFAR-100 | Clothing1M |
|---|---|---|---|
| # of train | 50,000 | 50,000 | 1,000,000 |
| # of test | 10,000 | 10,000 | 10,526 |
| # of classes | 10 | 100 | 14 |
| # of clients | 100 | 50 | 500 |
| Participation ratio | 0.1 | 0.1 | 0.02 |
| Model | ResNet-18 | ResNet-34 | ResNet-50$^{\dagger}$ |
| Learning rate | 0.03 | 0.01 | 0.003 |

### E.1    Heterogeneous data distribution settings on CIFAR-10

Figure 9 shows how we partition the CIFAR-10 dataset among the client's local datasets.

### E.2    Heterogeneous data distribution settings on CIFAR-100

Figure 10 shows how we partition the CIFAR-100 dataset among the client's local datasets.

### E.3    Noise transition matrix

Figure 11a and Figure 11b illustrate the practical transition matrices from our implementation, which show how labels are reassigned based on their ground truth labels.

### E.4    Pseudocode for FLR implementation

Algorithm 1 provides a detailed procedure for implementing the Federated Label-mixture Regularization (FLR) method, a strategy designed to tackle noisy labels within a FL environment.

The algorithm is structured into two distinct phases. The first phase serves as a warm-up round, applying the standard Federated Averaging (FedAvg) method utilizing standard Cross-Entropy (CE) loss. This phase forms a baseline global model trained on noisy labels. The main steps are as follows:

- **Initialization:** The server model parameters, denoted as $\theta_{server}$, are initialized with randomly chosen parameters, $\theta_0$.

- **Training iterations:** A loop for each epoch initiates, where a subset of clients $S^e$ are selected for training. For every chosen client, they start with a model whose parameters match those of the server model ($\theta_k = \theta_{server}$). The model parameters are updated through local gradient descent steps on their individual datasets, applying the standard CE loss.

- **Aggregation:** Lastly, the server gathers the client models into a global model by computing a weighted average. The weight for each client model is proportional to their dataset size.

In the second phase, the FedAvg procedure is augmented with the proposed FLR loss. The steps are akin to phase one, but with the following enhancements:

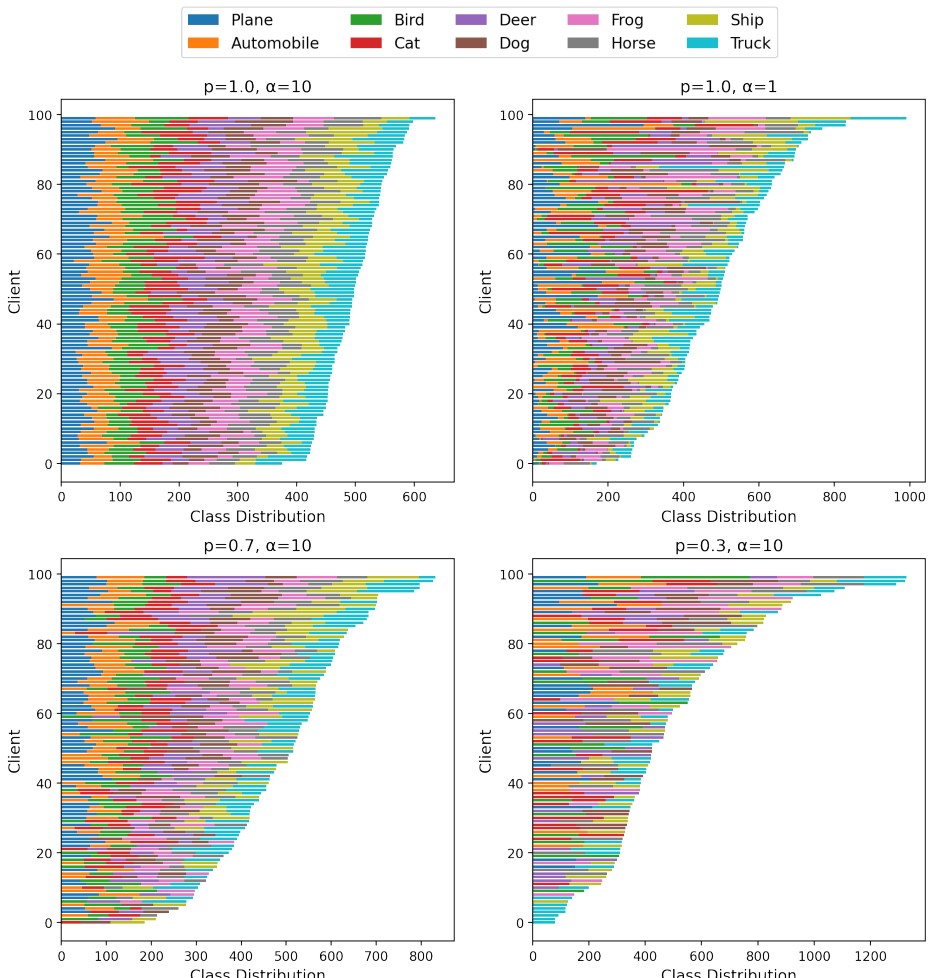

Figure 9: Heterogeneous data distribution settings on CIFAR-10.

- **Running average predictions:** For each mini-batch in a client's dataset, both global and local running average predictions are computed. The global running average predictions ($s_{k_i}$) incorporate the server model's predictions, whereas the local running average predictions ($m_{k_i}$) utilize the predictions of the client's own model.

- **Mixed labels:** A mixture label, $t_{k_i}$, is formed by combining both global and local running average predictions. This facilitates an effective correction of noisy labels.

- **Client update with FLR:** The client models are updated via gradient descent, applying the proposed FLR loss. The FLR loss consists of the standard CE loss and a regularization term, which is based on the distance between the current model's prediction and the mixture label.

The delineated algorithm offers a comprehensive blueprint for implementing FL in the presence of noisy labels, effectively handling data heterogeneity and varying label noise across different clients.

In addressing potential concerns regarding the computational demands posed by our FLR method, it is important to note that these are effectively managed through localized computations at the client level coupled with strategic synchronization. These strategies are integral to FL frameworks and are designed to manage the computational load efficiently. By employing localized computations and selective synchronization, we ensure that the additional complexities introduced by FLR do not overburden the system. This approach not only optimizes resource utilization but also confirms the practicality and feasibility of implementing FLR

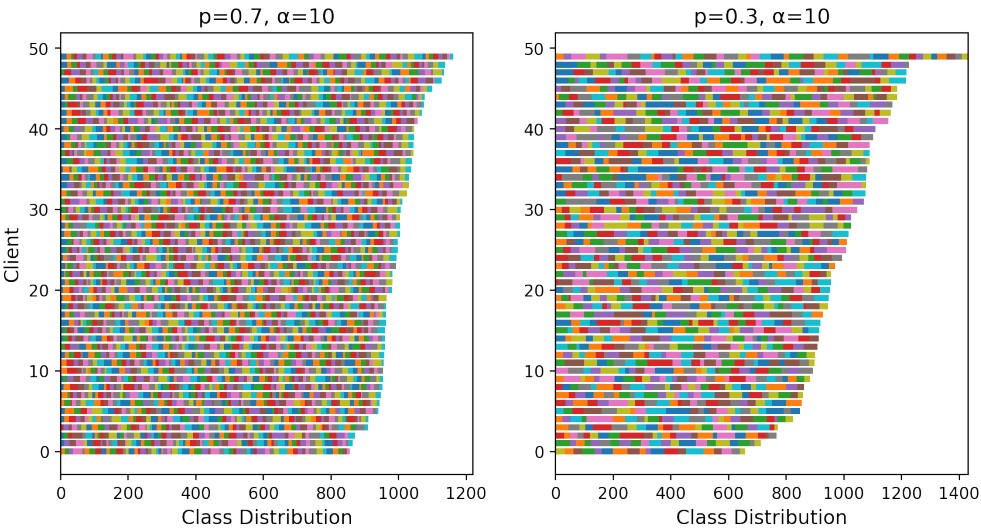

Figure 10: Data distributions of non-i.i.d. settings on CIFAR-100.

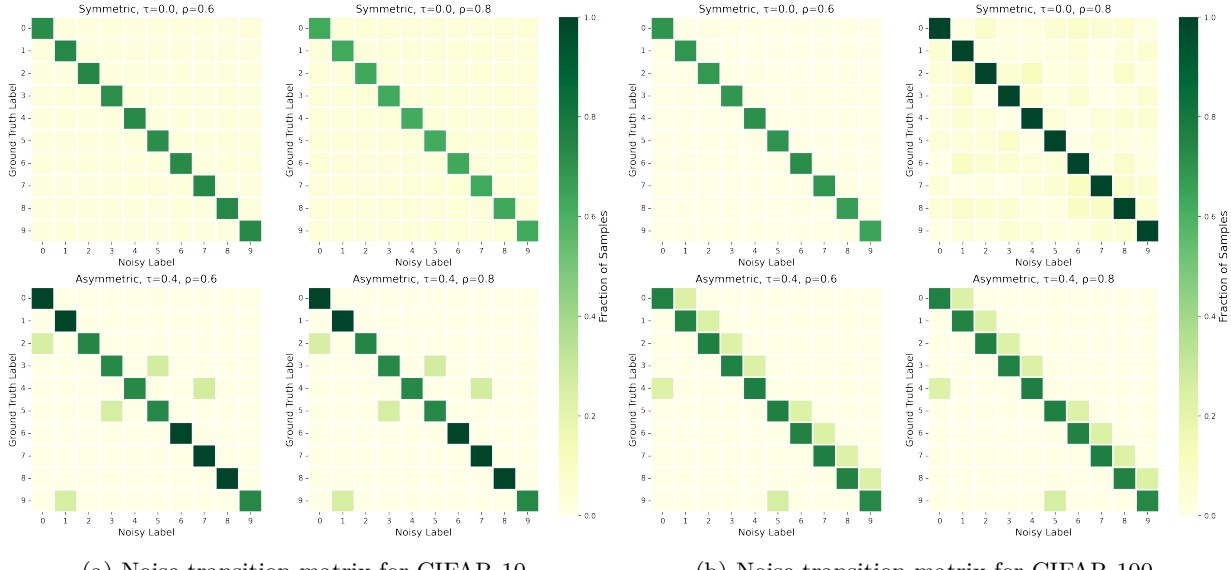

(a) Noise transition matrix for CIFAR-10.   (b) Noise transition matrix for CIFAR-100.

Figure 11: Noise transition matrices.

within the operational constraints typical of federated environments. By integrating these strategies, we maintain the method's efficiency, making it a viable solution for FL scenarios characterized by noisy labels.

### E.5 Implementation details

Our implementation details for the methods are as follows:

- FedProx (Li et al., 2020c) requires one coefficient $\mu$ for its proximal term. We chose $\mu = 0.001$ for our experiments.

- FedCorr (Xu et al., 2022) has a lot of hyperparameters. Among them, the number of rounds in finetuning and usual FL stage are set to be (300, 300) for CIFAR-10, (300, 400) for CIFAR-100, and (50, 50) for Clothing1M. We followed the values of original paper for the rest of hyperparameters.

- GCE (Zhang & Sabuncu, 2018) requires two hyperparameters, $q$ and $\kappa$. We selected $q = 0.7, \kappa = 0.1$ for CIFAR-10 and Clothing1M, and $q = 0.1, \kappa = 0.3$ for CIFAR-100 in our experiments.

- Co-teaching (Han et al., 2018) uses two models. In our implementation of Co-teaching for FL, we maintain two global models $w_1$ and $w_2$. When each round starts, both of them are given to each client as the local models $w_{1k}$ and $w_{2k}$, and each client trains them with Co-teaching process. After the local updates are done, the global models are updated by global aggregation respectively; $\{w_{1k}\} \to w_1$ and $\{w_{2k}\} \to w_2$.

- DivideMix (Li et al., 2020a) for FL is implemented in the same way as in the Co-teaching case. Regarding hyperparameters, we set 0.5 for the sharpening temperature, and 25 for the unsupervised loss coefficient.

- Early Learning Regularization (ELR) (Liu et al., 2020) requires two coefficients $\lambda$ and $\beta$ for its regularization term. After many trials of grid search, we found that $\lambda = 4.0$ with $\beta = 0.5$ for symmetric noise and $\beta = 0.9$ for asymmetric noise works better than any other hyperparameter values that we tested, in most of our experimental settings. We used $\lambda = 4.0$ and $\beta = 0.5$ also for Clothing1M.

- FLR require several coefficients $\lambda$, $\alpha$, $\beta$, and $\gamma$, and has the option of using linear scheduling for $\alpha$. $(\alpha, \beta, \gamma)$ is fixed to (0.9, 0.7, 0.5), and other settings are specified in Table 5.

- FLR$^+$ is used $\alpha_M = 1$ (beta distribution parameter used for Mixup augmentation) when coupled with FedCorr framework, and $\alpha_M = 4$ for DivideMix.

For learning strategy, we implement the training as follows:

$$\alpha(r) = \alpha \times \frac{r}{R}, \quad \beta(r) = \begin{cases} 0 & \text{if } r < \frac{R}{2} \\ \beta & \text{else} \end{cases}, \quad \gamma(r) = \begin{cases} 0 & \text{if } r < R_w \\ \gamma & \text{else} \end{cases}$$

where $R_w$ denotes the epoch for warmup training, which is set to 50. For the methods developed in CL settings such as Mixup (Zhang et al., 2018), GCE (Zhang & Sabuncu, 2018), Co-teaching (Han et al., 2018), DivideMix (Li et al., 2020a), and ELR (Liu et al., 2020), we just follow the implementation of their original papers.

Building on the equation 1, other regularizations are formulated according to the uses of $\alpha, \beta, \gamma$:

- Early Learning Regularization (ELR) Liu et al. (2020): This method is a specific case of FLR where $\alpha = 0, \gamma > 0$ in federated settings. The significant advantage of ELR is that it does not require additional storage for $\boldsymbol{s}_{k_i}$.

- Entropy Regularization (ER): In scenarios where $\alpha = \gamma = 0$, the regularization aligns closely with the entropy minimization term Grandvalet & Bengio (2004).

Table 5: Impelmentation details of FLR for our experiments.

| Dataset | Case | $\lambda$ |
|---|---|---|
| CIFAR-10 (CIFAR-10N) | Symmetric noise | 2.0 |
| | Asymmetric noise | 3.0 |
| | Symmetric noise, w/ FedCorr | 3.0 |
| | Asymmetric noise, w/ FedCorr | 3.0 |
| CIFAR-100 | Symmetric noise | 2.5 |
| | Asymmetric noise | 5.0 |
| | w/ Mixup | 2.0 |
| | w/ FedCorr | 5.0 |
| Clothing1M | - | 2.0 |
| | w/ FedCorr | 3.0 |

- Server-aware Learning Regularization (SLR): This is another special case of FLR, specifically when $\alpha = 1$, where local predictions are directly penalized by global predictions. Although similar to general distillation, SLR differs in its use of the logarithmic function rather than the Kullback-Leibler function (Hinton et al., 2015). In FL, other methods like FedProx (Li et al., 2020c) and FedDyn (Durmus et al., 2021) improve generalization against data heterogeneity by reducing weight divergence between the local and global model. However, our approach is distinct in that it directly mitigates divergence between predictions, rather than model parameters.

### E.6 Comparison with other label regularization methods

Our adaptive regularization, especially ER, slightly overlaps with calibration methods, which have been extensively studied in CL. We compare ER with some confidence-related previous methods:

- Entropy Minimization (EM) (Grandvalet & Bengio, 2004) requires one coefficient for its regularization term. We set the coefficient to be 4.

- Label Smoothing (LS) (Pereyra et al., 2017) and Negative Label Smoothing (NLS) Wei et al. (2022) requires one smoothing parameter. We set its value to be 0.1 for LS, and -0.05 for NLS.

- Temperature Scaling (TS) (Guo et al., 2017) requires sharpening temperature. We set the value to be 4.0.

- Entropy Regularization (ER): In scenarios where $\alpha = \gamma = 0$, the regularization aligns closely with the entropy minimization term (Grandvalet & Bengio, 2004).

Table 6 shows the results of ER comparing with Entropy Minimization (EM) (Grandvalet & Bengio, 2004), Label Smoothing (LS) (Pereyra et al., 2017), Negative Label Smoothing (NLS) (Wei et al., 2022), and Temperature Scaling (TS) (Guo et al., 2017). It is observed that even simple entropy penalization for each instance with either EM or ER can make the model to be robust towards noisy labels.

## F  Gradients of the adaptive regularization

**Lemma.** (Gradient of the regularization). The gradient of the loss is defined in equation 2. Similar to Liu et al. (2020), we denote the federated adaptive regularization by

$$R(\theta) = \log\left(1 - \langle \boldsymbol{p}_{k_i} \cdot \boldsymbol{t}_{k_i} \rangle\right) \tag{3}$$

Table 6: Comparison with other label regularization methods, on CIFAR-10 with i.i.d. setting.

| Method | Symmetric | | Asymmetric | |
| | $\tau = 0.0$ | | $\tau = 0.4$ | |
| | $\rho = 0.6$ | $\rho = 0.8$ | $\rho = 0.6$ | $\rho = 0.8$ |
|---|---|---|---|---|
| CE | 74.58±1.00 | 67.76±1.14 | 85.90±0.25 | 83.13±0.35 |
| EM | 86.14±0.45 | 82.90±1.01 | 88.25±0.20 | 86.17±0.69 |
| LS | 73.95±0.79 | 67.42±0.67 | 84.04±0.12 | 80.23±0.58 |
| NLS | 79.55±0.87 | 71.36±0.64 | 68.86±0.62 | 66.87±0.87 |
| TS | 82.28±0.65 | 76.38±1.34 | 87.45±0.34 | 84.97±0.63 |
| ER | **86.45±0.45** | **83.12±0.88** | **88.73±0.24** | **86.35±0.42** |

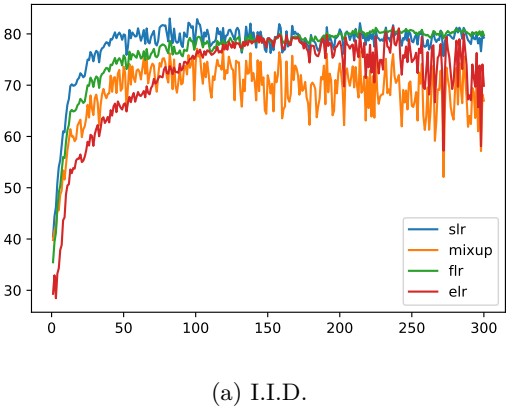

(a) I.I.D.

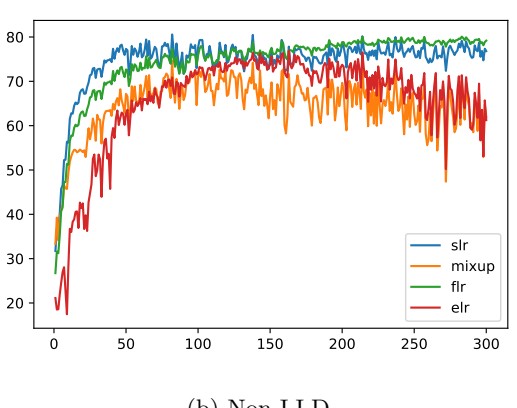

(b) Non I.I.D.

Figure 12: (a) CIFAR-10 of the i.i.d. setting, with symmetric noise of $(\rho, \tau) = (1.0, 0.0)$, (b) CIFAR-10 of the non-i.i.d. setting, with symmetric noise of $(\rho, \tau) = (1.0, 0.0)$

The gradient of $R$ is

$$\nabla R(\theta) = \frac{\nabla \left(1 - \langle \boldsymbol{p}_{k_i} \cdot \boldsymbol{t}_{k_i} \rangle \right)}{1 - \langle \boldsymbol{p}_{k_i} \cdot \boldsymbol{t}_{k_i} \rangle} \tag{4}$$

It can be expressed with the probability estimate in terms of the softmax function and the deep-learning mapping $N_x(\theta)$, $\boldsymbol{p}_{k_i} = \frac{\exp(\mathbf{N_x}(\theta)_{\mathbf{c}})}{\sum_c \exp(N_x(\theta)_c)}$, where the bolded numerator is a softmax probability vector. Substituting and solving this, we get

$$\nabla R(\theta) = \frac{-\nabla N_x(\theta)}{1 - \langle \boldsymbol{p}_{k_i} \cdot \boldsymbol{t}_{k_i} \rangle}(\boldsymbol{p}_{k_i} \odot \boldsymbol{t}_{k_i} - \langle \boldsymbol{p}_{k_i} \cdot \boldsymbol{t}_{k_i} \rangle \cdot \boldsymbol{p}_{k_i})$$

$$= \frac{-\nabla N_x(\theta)}{1 - \langle \boldsymbol{p}_{k_i} \cdot \boldsymbol{t}_{k_i} \rangle} \begin{bmatrix} \boldsymbol{p}_{k_i}^{(1)} \cdot \sum_{r=1}^{C}(\boldsymbol{t}_{k_i}^{(r)} - \boldsymbol{t}_{k_i}^{(1)})\boldsymbol{p}_{k_i}^{(r)} \\ \vdots \\ \boldsymbol{p}_{k_i}^{(C)} \cdot \sum_{r=1}^{C}(\boldsymbol{t}_{k_i}^{(r)} - \boldsymbol{t}_{k_i}^{(C)})\boldsymbol{p}_{k_i}^{(r)} \end{bmatrix} \tag{5}$$

## G  Learning curves across different data heterogeneity

Building on the equation 1, other regularizations are formulated according to the uses of $\alpha, \beta, \gamma$:

- ELR (Liu et al., 2020): This method is a specific case of FLR where $\alpha = 0, \gamma > 0$ in federated settings. The significant advantage of ELR is that it does not require additional storage for $\boldsymbol{s}_{k_i}$.

- SLR: This is another special case of FLR, specifically when $\alpha = 1$, where local predictions are directly penalized by global predictions. Although similar to general distillation, SLR differs in its use of the logarithmic function rather than the Kullback-Leibler function (Hinton et al., 2015). In FL, other methods like FedProx (Li et al., 2020c) and FedDyn (Durmus et al., 2021) improve generalization against data heterogeneity by reducing weight divergence between the local and global model. However, our approach is distinct in that it directly mitigates divergence between predictions, rather than model parameters.

In this section, we present the learning curves over 300 epochs for four different strategies: ELR, SLR, Mixup, and FLR. The curves are plotted under both the i.i.d. and non-i.i.d. settings.

Our observations indicate that both FLR and SLR generally perform well, with FLR consistently exhibiting superior performance (Figure 12). This underlines the effectiveness of our proposed FLR method in combating the effects of noisy labels in the FL environment.

On the other side, as the number of epochs increases, ELR and Mixup's performance tends to deteriorate. This is particularly evident in their generalization capabilities, which decrease due to memorization. This memorization effect is particularly evident when these methods are exposed to prolonged training, which is a common situation in a real-world scenario.

