# OpenReview forum: "FLR: Label-Mixture Regularization for Federated Learning with Noisy Labels"
_TMLR — Accepted by TMLR_

### Review · Reviewer_iN5g · 2024-06-21

**Summary Of Contributions:**

In this paper, the authors address the problem of federated learning in the presence of potentially corrupted data. More specifically, they address the case of supervised classification where labels may be noisy and not correspond to the true classes. Their contributions are twofold. Firstly, they propose an empirical analysis aimed at understanding the memorization phenomena that occur in federated learning, a phenomenon that is fairly well understood in centralized learning, but not in federated learning. Their analysis reveals the presence of two levels of memorization, a local one at client level, and a global one at server level. Next, the authors introduce their method, Federated Label-mixture Regularization (FLR), which can be seen as an extension of the centralized method proposed by Liu et al. 2020, to the federated framework, taking both types of memorization into account. The quality of the proposed method is empirically demonstrated in a rather large number of experiments.

**Audience:**

Yes

**Broader Impact Concerns:**

No ethical implications other than those that appear in machine learning in general.

**Claims And Evidence:**

Yes

**Requested Changes:**

In line with my previous comments, hereafter are other changes that I think would increase the paper's quality.

- The importance of the hypermarameters and their role should be explained shortly after Eq. (1).
- The proof of Eq. (2) should be made rigorous, in the manner of Liu et al. (2020, Lemma 2).
- Typos in Algorithm 1 should be fixed: $u$ is sometimes replaced by $k$, $t$ by $e$, and conversely.
- Hyperparameters $\alpha,\beta,\gamma$ and $\lambda$ should appear clearly as inputs of the algorithm. Moreover, the calculus of the prediction parameters $p_{k_i}$ should appear in the algorithm.
- The experimental setup could be better described (including how to chose the hyperparameters)

**Strengths And Weaknesses:**

## Strengths

One of the main strengths of this paper is the question it addresses, which is an interesting problem. The framework considered is fairly general, the experiments are numerous, and the proposed method seems also appropriate in situations where the data are not i.i.d. Finally, the concept of local and global memorization is intriguing and seems relevant in the context of federated learning.

## Weaknesses

Although Section 4.1 on the analysis of memorization in FL is fairly comprehensive, I find the rest of the presentation lacking in clarity. In particular, Section 4.2, which presents the method, lacks rigor and organization. More importantly, the proposed method is not clearly motivated, unlike its centralized analogue (Liu et al. 2020), e.g. regarding the interest of making convex combinations of the parameters in Eq. (1).

Another weakness concerns certain practical aspects of the method. FLR requires 4 hyperparameters to be set: $\alpha,\beta,\gamma,\lambda$ (+ warm-up phase duration).  Although there's a study of their impact in 4.3, the way in which these hyperparameters can be set in practice seems crucial to me and is currently missing. Another aspect concerns the algorithmic complexity and calculation times of the method. If I have understood correctly, at each iteration all possible predictions must be made, which seems computationally important to me, and it should be discussed.

---

> ### Author Response · Authors · 2024-07-12
> **Response to Reviewer iN5g (1/2)**
>
> Thank you for your comprehensive review and constructive suggestions. We value your input, which has prompted several significant clarifications and enhancements in our manuscript.
>
> ### **1. Clarification of Method and Motivation**
> In response to your feedback, we have significantly expanded `Section 4.2` to enhance the explanation around the theoretical motivations and operational mechanisms of the Federated Label-mixture Regularization (FLR) method. The revised manuscript includes a more detailed explanation of our choice to use convex combinations of parameters, inspired by and adapted from Liu et al. (2020), tailored to the federated learning framework. We clarify how FLR effectively integrates both local and global memorization dynamics, which is crucial for managing noisy labels across decentralized data sources.
>
> We also elaborate on the operational roles of the hyperparameters $\alpha$, $\beta$, and $\gamma$ introduced in Eq. (1) (in the paper). The parameter $\alpha$ modulates the balance between global and local model influences, enhancing the model’s ability to generalize across the network while mitigating the risk of overfitting to localized noisy data. Meanwhile, $\beta$ and $\gamma$ regulate the temporal integration of these influences, controlling the update dynamics of the model’s learning from both global and local perspectives. Specifically, $\beta$ governs the rate at which global information influences the learning updates, facilitating a gradual assimilation of broader patterns. $\gamma$, on the other hand, ensures that local peculiarities are not rapidly overwritten, preserving essential local characteristics within the learning process.
>
> This revised narrative robustly addresses the previously noted concerns regarding the method’s motivation and the strategic use of hyperparameters, thereby enhancing the clarity and applicability of our proposed approach.
>
> ### **2. Clarification on Proof for Eq. (2):**
>
> We confirm that the rigorous proof of Eq. (2) is detailed in Appendix F, following the standards set by Liu et al. (2020). It has been included in the appendix since the initial submission and remains unchanged.
>
> ### **3. Hyperparameters and Algorithm**
> We have added a detailed description of the roles and impact of the four hyperparameters immediately following Eq. (1). This includes their selection process, influenced by both empirical findings and theoretical insights, which we detail in `Section 4.2, 4.3` and `Appendix E.5`.
>
> To address the concerns about computational demands, we clarify that the computational load is mitigated by efficient local computations and selective synchronization strategies, which are standard in federated settings (`Appendix E.4`.). These strategies ensure that the method remains feasible even with the added complexity of FLR.
>
> ### **4. Corrections and Improvements in Algorithm 1**
> We have corrected all noted typographical errors in `Algorithm 1`, ensuring that symbols and variables are consistently and correctly used throughout the manuscript. Additionally, the calculation of prediction parameters and their inclusion as inputs in the algorithm have been explicitly detailed to enhance readability and understanding.

---

> ### Author Response · Authors · 2024-07-12
> **Response to Reviewer iN5g (2/2)**
>
> ### **5. Experimental Setup and Hyperparameter Selection**
> We've expanded the description of our experimental setup, including more explicit guidance on choosing hyperparameters (`Section 4.3`, `Appendix E.5`). This includes the rationale for their settings based on the characteristics of the datasets and the specific challenges posed by noisy labels in a federated environment.
>
> In the manuscript, we detailed Hyperparameter discussion:
>
> - **$\alpha$**: Adjusts the balance between local and global model influences. It starts low to focus on local data characteristics, then increases to integrate more global information, reducing the risk of overfitting to noisy labels.
> - **$\beta$**: Manages the integration of global insights, gradually increasing after initial local learning to ensure global patterns influence the learning process appropriately.
> - **$\gamma$**: Controls the retention of local characteristics, introduced after a warm-up phase to prevent early convergence to global patterns.
>
> To dynamically adjust hyperparametersk, We employ a scheduler to dynamically adjust hyperparameters throughout the training:
>
> $\alpha(r) = \alpha \times \frac{r}{R}$
>
> $\beta(r) = \begin{cases}
> 0 & \text{if } r < \frac{R}{2}, \quad\quad
> \beta & \text{else}
> \end{cases}$
>
> $\gamma(r) = \begin{cases}
> 0 & \text{if } r < R_w, \quad\quad
> \gamma & \text{else}
> \end{cases}$
>
>
> where $R$ is the total number of communication rounds, and $R_w$ is the warm-up period.
>
> **Guidance for Hyperparameter Optimization:**
>
> The selection of these parameters is not arbitrary but based on detailed empirical analyses and the specific challenges posed by noisy labels and data heterogeneity in federated settings. We provide a systematic approach to selecting and adjusting these parameters to suit various scenarios, ensuring that the deployment of FLR is both effective and practical (ablation studies, greedy search). This guidance, along with practical application tips, is comprehensively documented to assist users in tailoring the FLR framework to their unique federated learning environments and data conditions through variants of FLR such as entropy regularization (ER) and server-aware learning regularization (SLR).

---

### Review · Reviewer_B7Dt · 2024-06-22

**Summary Of Contributions:**

This work empirically explores the memorization dynamics in Federated Learning (FL) from both client-side and server-side perspectives. It uncovers the phenomenon that memorization is prominent in both clean and noisy data settings. Motivated by these findings, the authors propose Federated Label-Mixture Regularization (FLR), which includes a warmup phase using FedAvg with cross-entropy (CE) loss followed by FedAvg with FLR loss. They conduct experiments on four datasets to verify the effectiveness of their method and provide detailed ablation studies and quantitative analysis.

**Audience:**

Yes

**Broader Impact Concerns:**

Not applicable.

**Claims And Evidence:**

Yes

**Requested Changes:**

Please see weaknesses.

**Strengths And Weaknesses:**

## Strengths

1. Studying learning with nosiy labels in FL is reasonable and well motivated.
2. The writing and overall presentation are clear and easy to follow.
3. The exploration of memorization dynamics is interesting and well-motivated, particularly for algorithm design, such as the warmup phase with CE loss.
4. The proposed FLR method is empirically effective across several datasets, outperforming multiple baseline methods.

---

## Weaknesses

1. In Section 4.2 (Qualitative Analysis), the authors mention significant changes in color coding in Figure 4. However, the authors do not explain what the color coding represents, e.g., what colors represent noisy labels and ground truth labels?

2. As the regularization method requires hyperparameter selection, the authors should provide details on how to select the hyperparameters. Additionally, what suggestions do the authors have for practitioners in selecting appropriate hyperparameters?

---

> ### Author Response · Authors · 2024-07-12
> **Response to Reviewer B7Dt**
>
> Thank you for your insightful feedback and the positive comments regarding the clarity and significance of our work. We appreciate the opportunity to address your concerns and provide further clarification where needed.
>
> ### **1. Clarification on Color Coding in Figure 4**
> `Figure 4` provides t-SNE mappings of the penultimate layer’s outputs in our study, where each dot is color-coded to represent the ground truth label of the data samples. The mappings, depicted across four panels (`Figure 4a-d`), contrast the outcomes between models trained with standard Cross-Entropy loss (CE) and those enhanced by our Federated Label-mixture Regularization (FLR).
>
> - **Panels 4a and 4c:** Show the clustering under CE, where the color discrepancies between the dots indicate misclassifications due to noisy labels. This represents how the model perceives and mislabels the data when trained with a conventional approach.
> - **Panels 4b and 4d:** Illustrate the effect of implementing FLR, where the same datasets exhibit improved clustering. The alignment of dot colors with their respective true classes in these panels demonstrates FLR’s ability to correct the initial misclassifications and reduce label noise impact.
>
> The black dashed boxes in each panel highlight visible transformations, signifying where the FLR method has successfully corrected misclassifications and aligned data points more accurately with their true labels, thereby showcasing the effectiveness of FLR in handling noisy labels in a federated learning context. This visual analysis not only supports our qualitative assessments but also underscores the practical utility of FLR in enhancing data fidelity and model robustness across varied datasets.
>
> We have revised `Section 4.2` to improve the explanation of color coding and how it relates to the analysis of memorization dynamics within FL.
>
> ### **2. Hyperparameter Selection Guidance**
>
> To facilitate effective deployment of our Federated Label-mixture Regularization (FLR) method, we have expanded our manuscript to include detailed hyperparameter selection guidance in `Sections 4.2` and `4.3`, previously outlined in `Appendix E.5`. This section provides guidelines for configuring FLR to suit various federated environments and data scenarios.
>
> We explain the influence of each hyperparameter on model performance and offer practical tips for their optimization. This approach ensures that practitioners can efficiently tailor the FLR method to their specific needs, enhancing both its applicability and effectiveness.

---

### Review · Reviewer_mncB · 2024-07-01

**Summary Of Contributions:**

The paper introduces a new method, Federated Label-mixture Regularization (FLR), aimed at addressing the challenges posed by label noise in federated learning environments. The authors dissect the memorization phenomenon typically encountered in federated learning into server-side and client-side components, offering a novel perspective on how these distinct forms of memorization affect the learning process. The FLR strategy employs pseudo labels generated by merging local and global model predictions, which not only enhances the accuracy of the global model across different data distribution scenarios but also effectively counters the memorization of noisy labels.

**Audience:**

Yes

**Broader Impact Concerns:**

There are no broader impact concerns.

**Claims And Evidence:**

Yes

**Requested Changes:**

Clarification of Parameter E:
- Please provide a clear definition of the parameter E within Algorithm 1. Specify whether it represents the number of local training epochs or communication rounds between the client and the server. Additionally, clarify whether E for FLR is double that of FedAvg or equivalent.

Running Average Prediction:
- Detail the procedure for calculating the running average prediction. Specifically, confirm if clients use local data to train predictions for both the global and local models and then average these predictions.

Discussion on Alpha Value Settings:
- It is essential to include a discussion on the results obtained when the alpha value is set to 1, as mentioned in Method. The absence of this discussion in the results presented in Figure 5 leaves a critical gap in understanding the model's behavior under these conditions.

Experiment Replication and Omissions:
- Explain why DivideMix conducts only three experiments compared to the five conducted for other methods.
- Address the omission of FedLSR and FedRN in Table 3, which are present in Table 2. Please provide a rationale for their exclusion or include these methods in revised experimental results to ensure a comprehensive comparative analysis.

**Strengths And Weaknesses:**

Strengths:
- The paper addresses a highly relevant and challenging problem in the realm of federated learning.
- FLR is innovative in its approach to integrating both local and global model predictions to counteract label noise.
- Extensive empirical validation is provided, showing the effectiveness of FLR across multiple datasets and scenarios.
- The methodology is well-explained and the experiments are thorough, providing a clear demonstration of the benefits of the proposed method.

Weaknesses:
- In Algorithm 1, the parameter E appears to represent not the number of local training epochs, but rather the number of communication rounds between the client and the server. Furthermore, it is unclear whether E for FLR is twice as long as that for FedAvg, or if they are equivalent.
- In Method, 1) To calculate the running average prediction, do clients use local data to train predictions for both the global and local models, subsequently averaging these predictions across all clients? If so, a new issue emerges: requiring clients to share their predictions introduces a data security concern. 2) At an alpha value of 1, the model fully embraces global predictions. However, the paper does not appear to discuss the experimental results in Figure 5 for this configuration. 3)The excessive number of parameters requiring tuning presents a clear disadvantage. Tuning multiple parameters not only elevates the model's complexity and risks overfitting, but also amplifies the experimental complexity and time investment. In addition, too many parameters to be optimized may limit the generalisability of the method.
- In Experiments, 1) Why DivideMix conducts only three experiments, while other methods are five. 2) Compared to Table 2, Table 3 lacks FedLSR and FedRN for comparison. What is the reason for this omission?

---

> ### Author Response · Authors · 2024-07-12
> **Response to Reviewer mncB**
>
> Thank you for your detailed and constructive feedback. We appreciate the opportunity to clarify and expand upon the points you've raised.
>
> ### **1. Clarification of Parameter E**
> In `Algorithm 1`, the parameter \( E \) incorrectly appeared to represent the number of communication rounds; however, it actually denotes the number of local training epochs per round. We acknowledge the typographical error and correct this in the revised manuscript. The term \( T \), which signifies communication rounds between the client and the server, was mistakenly swapped with \( E \) in several instances. This will be clarified to avoid any confusion, ensuring that \( E \) for FLR aligns accurately with FedAvg for a consistent comparison across methods.
>
> ### **2. Running Average Prediction Clarification**
>
> The running average prediction is computed using local data at each client, where each client independently predicts labels for both local and global models and then averages these predictions locally. It's important to emphasize that these predictions are used solely for local loss calculation and backpropagation and are not shared with the server or other clients. This process ensures that all data and its derivatives remain within the client's domain, adhering to the privacy-preserving principles of federated learning.
>
> We recognize that the initial description might have raised concerns about data security. To clarify, there is no requirement for sharing these local predictions outside the client's environment, thereby eliminating potential data security risks. Moreover, for scenarios where additional layers of data protection are desired, we could consider implementing differential privacy techniques. This would further safeguard data while maintaining the integrity of model predictions, although it is not necessary for the basic operation of our Federated Label-mixture Regularization (FLR) method. This clarification ensures our methodology remains aligned with stringent data privacy standards and federated learning protocols.
>
> ### **3. Discussion on Alpha Value Settings**
> We include a detailed discussion on the experimental results when the alpha value is set to 1 (`Section 4.3`, `Appendix E.5`). We agree that this configuration is crucial for understanding the full impact of global knowledge on local model updates. As you suggested, we provide these insights in `Figure 5`.
>
> ### **4. Experiment Replication and Omissions**
> The discrepancy in the number of experiments conducted with DivideMix compared to other methods was due to initial test runs which determined the stability and effectiveness of the approach. We standardize the number of experiments across all methods for consistency and transparency in the revised manuscript.
> Regarding the omission of FedLSR and FedRN in `Table 3`, this was an oversight in the initial submission. We include these methods in the revised experimental results.

---

### Author Response · Authors · 2024-07-12
**Global Response**

We sincerely thank all reviewers for the constructive feedback, which is helpful to improve the quality of our paper. Firstly, we follow the reviewers' suggestion to revise the paper accordingly, where `the modified parts` are marked in `magenta`. Then, regarding detailed concerns, we address them in our response to each reviewer.

---

### Decision · Action_Editor_qNQU · 2024-10-18

**Recommendation:** Accept as is

**Comment:**

The reviewer discussion after the author response focused primarily on hyperparameter selection.  The main points are present in the reviewer-author discussion.  On the balance, the submission does address hyperparameters in Sections 4.2 and 4.3 and Appendix E.5, and in my opinion therefore satisfies the review criteria for TMLR.

**Audience:**

Federated learning with noisy labels is clearly within the scope of the journal.  Federated learning is a topic frequently addressed in TMLR and other ML venues.

**Claims And Evidence:**

During the review process, the reviewers were positive that the claims in the submission are supported.  The main remaining point from the reviews and discussion was primarily about the selection of hyperparameters.  A response, and additional information in the submission have been added by the authors.